# Drought Early Warning in Agri-Food Systems

**Maarten van Ginkel \* and Chandrashekhar Biradar**

International Center for Agricultural Research in the Dry Areas (ICARDA), Beirut 1001, Lebanon;
c.biradar@cgiar.org
\* Correspondence: maartenvanginkel1951@gmail.com

**Abstract:** Droughts will increase in frequency, intensity, duration, and spread under climate change. Drought affects numerous sectors in society and the natural environment, including short-term reduced crop production, social conflict over water allocation, severe outmigration, and eventual famine. Early action can prevent escalation of impacts, requiring drought early warning systems (DEWSs) that give current assessments and sufficient notice for active risk management. While most droughts are relatively slow in onset, often resulting in late responses, flash droughts are becoming more frequent, and their sudden onset poses challenging demands on DEWSs for timely communication. We examine several DEWSs at global, regional, and national scales, with a special emphasis on agri-food systems. Many of these have been successful, such as some of the responses to 2015–2017 droughts in Africa and Latin America. Successful examples show that early involvement of stakeholders, from DEWS development to implementation, is crucial. In addition, regional and global cooperation can cross-fertilize with new ideas, reduce reaction time, and raise efficiency. Broadening partnerships also includes recruiting citizen science and including seemingly subjective indigenous knowledge that can improve monitoring, data collection, and uptake of response measures. More precise and more useful DEWSs in agri-food systems will prove even more cost-effective in averting the need for emergency responses, improving global food security.

**Keywords:** drought; drought early warning systems; climate change; mitigation; food security

## 1. Introduction

Two comprehensive reviews on drought early warning systems (DEWSs) have been published in the past decade [1,2]. In their review of more than 20 DEWSs, Pulwarty and Sivakumar [1] stress that a DEWS is more than a mere forecast of oncoming stress, but includes ongoing monitoring, risk assessment, and decision support information involving many sectors and disciplines within society, integrated into an effective communication package that engages communities in a timely fashion. Recent deadly floods in parts of northwestern Europe and the heat and associated fires in northwest USA and southwest Canada in mid-2021 remind us that effectively integrated and well-communicated early warning systems remain a topic of high priority everywhere. The DEWSs reviewed highlight in particular the lack of learning links between local, national, and international DEWS efforts. To improve the decision-making process, the authors propose a formal information pedigree with reliable, accurate, and precise information specifically identifying, for users, the danger levels regarding production impacts, so that all will realize the economic and social benefits of DEWSs and support action. This requires strong leadership with an active attitude to droughts, which often have a slow onset that results in a delayed, reactive response. DEWSs also allow communities to prepare ahead of stress, taking decisions on measures that bring about long-term risk reduction by linking a crisis calendar to a decision calendar. Successful implementation of such preparedness will require educating and training actors at all levels, stressing interlinkages among risk assessment, decision-making, communication, and responses. The review presents the various

steps involved in achieving these outcomes, many of which are also relevant to other stress early warning systems beyond drought.

Funk et al. (2019) reviewed 30 years of DEWSs, focusing on famine avoidance as we strive for global food security [2] They stressed how DEWSs allow for finite resources to be implemented most effectively in times of drought, making the largest positive impact. They focused on the Famine Early Warning Systems Network DEWS, from its inception following the mid-1980s droughts in the Sahel, in particular describing how food insecurities are anticipated, quickly identified, targeted, and responded to. Drought indicators and expert judgements are both needed, with the network using a scenario development process to draw on a wide group of scientists around the world. Food security experts then use the information to develop alerts by country, making use of monitoring, field assessment, and prediction. The review draws lessons from relatively effective and ineffective responses to droughts in the horn of Africa (especially Kenya, Ethiopia, and Somalia) from 2011 to 2017. Factors include ongoing conflict in some countries obstructing relief, the need to interlink international partners, and improvements needed in monitoring and modeling. The review outlines actions to be taken before the season, such as canvassing historical information guidance, and during the season with monitoring tools (including satellite-based systems), as well as land-based hydrological parameters, in order to accurately quantify relative water availability for crops, pastures, and people. Funk et al. (2019) concluded by stressing that DEWSs are in need of continuous improvements or "science in action" to provide humanitarian responses. [2]

The present paper focuses on recent advances in DEWSs, building on these two reviews and other relevant publications, focusing on how DEWSs can help prepare agrifood systems to respond to increasing drought frequency and severity. We also stress the role of human activities in creating conditions for drought to occur, and how, by contrast, involving stakeholders and their knowledge along the entire impact pathway can make DEWSs even more effective in addressing and avoiding droughts. We believe these new points will enrich the debate and serve the DEWS community and beyond to its beneficiaries.

## 2. Rationale of Drought Early Warning Systems

Droughts impact many sectors in society, both simultaneously and sequentially, and they are considered among the costliest natural disasters [3]. Many of the years in the present century when world grain production fell below consumption requirements were also years of severe droughts [4]. The Intergovernmental Panel on Climate Change (IPCC) has, in every report, predicted further drought events, especially in lower latitudes. Because drought can severely limit crop growth and, hence, food security, proactive risk management approaches are called for, including DEWSs to move away from the reactive, crisis-management methods of the past [5]. DEWSs will facilitate targeted emergency food aid but should also be used to develop long-term plans and measures for sustainable water resource management into the future, to help avoid drought disaster reoccurrence [6].

Priorities to address drought were updated and agreed in the Sendai Framework for Disaster Reduction, 2015–2030, as follows [7]:

Priority 1: Understanding disaster risk.
Priority 2: Strengthening disaster risk governance to manage disaster risk.
Priority 3: Investing in disaster risk reduction for resilience.
Priority 4: Enhancing disaster preparedness for effective response and to "Build Back Better" in recovery, rehabilitation, and reconstruction.

Disaster risk is here defined as per UNISDR (2009): "The potential disaster losses, in lives, health status, livelihoods, assets and services, which could occur to a particular community or a society over some specified future time period" [8].

The Sendai Framework contains seven global targets, one of which is Early Warning Systems: "Substantially increase the availability of and access to multi-hazard early

warning systems and disaster risk information and assessments to people by 2030" [7] (p. 12). Hazard is "a potentially damaging physical event, phenomenon or human activity that may cause the loss of life or injury, property damage, social and economic disruption or environmental degradation. Hazards can include latent conditions that may represent future threats and can have different origins: natural (geological, hydrometeorological and biological) or induced by human processes (environmental degradation and technological hazards)" [7].

The Integrated Drought Management Programme (IDMP), sponsored by the World Meteorological Organization (WMO) and the Global Water Partnership (GWP), includes three pillars for a successful drought policy focused on risk reduction [9]:

1. Comprehensive monitoring and early warning system.
2. Vulnerability and impact assessment.
3. Mitigation, preparedness, and response.

The IDMP identifies four interrelated key elements needed in DEWSs:

1. Disaster risk knowledge based on the systematic collection of data and disaster risk assessments.
2. Detection, monitoring, analysis, and forecasting of the hazards and possible consequences.
3. Dissemination and communication of authoritative, timely, accurate, and actionable warnings and associated information on likelihood and impact by an official source.
4. Preparedness at all levels to respond to the warnings received.

Vulnerability is "the conditions determined by physical, social, economic, and environmental factors or processes, which increase the susceptibility of a community to the impact of hazards" [10].

One of the first international meetings aimed at developing joint DEWS expertise was the Expert Group Meeting held in Lisbon, Portugal, 5–7 September 2000. From the start, user needs were emphasized, as was the necessity to address environmental, economic, and social impacts [11]. These international interactions eventually culminated in the WMO developing a stepwise Multi-Hazard Early Warning System checklist for easy use [12].

Modern DEWSs cannot be considered without gender, because there are gender differences in how food insecurity and attempts to achieve it are experienced, with women and girls often disadvantaged [13,14]. Women, girls, and youth are best engaged from the early planning stages onward to give their needs full consideration, which is essential for uptake and community-wide implementation of any early warning system [14]. An additional risk of neglecting gender inequalities in agri-food systems is that, if on-farm crop yields are lost due to prolonged drought, children may be forced to drop out of school to earn additional income [14].

As the most food-insecure part of the population will suffer most, DEWSs should be prioritized to monitor their situation first and foremost, allowing for rapid well-targeted deployment of often limited remedial resources [2]. Given that the most food-insecure people tend to live in remote rural areas with limited infrastructure, remote sensing approaches to monitoring are becoming more important. Increasingly, field-based survey techniques are being complemented by remote sensing and GIS approaches to improve drought monitoring and expand scale efficiently [6]. While remote sensing has increased area covered and response speed, some drawbacks include the continued requirement of ground truthing of the algorithms, which, due to costs and time needed, may be declining in frequency and area covered. In much of the developing world, small farm size is another challenge for remote sensing technologies to monitor developments on that smaller scale, and for analytical tools to interpret data in a fashion relevant to smallholder farming communities.

## 3. DEWSs within Food-System Transformation

Agri-food systems today are failing to provide sufficient food of adequate nutritional quality, benefitting everyone equally, equitably, and ethically, and they often have a negative impact on the environment, exacerbated by climate change [15,16]. By 2015, global food systems contributed 34% of all anthropogenic greenhouse gas emissions from production, processing, transport, packaging, and consumption. Africa and Latin America are the highest-emitting regions, with their food systems responsible for two-thirds of their total emissions, but as national economies become more energy-intensive, the share of food system emissions is dropping [17].

Resilience to climate shocks is one the top four goals to improve food systems [15]. The Covid-19 pandemic has further aggravated and shone a light on the fragilities in existing food systems, especially in the developing world [15,18]. Labor-intensive production, access, and transport issues, internal and external, have been affected as a result of shortage in seasonal and migrant workers. Perishable food commodities, such as vegetables, fruit, meat, and dairy, have been particularly hard hit [18,19].

Agri-food systems must transform in order to meet the Sustainable Development Goals and food and nutrition security beyond 2030. An extensive study and stakeholder consultation by the CGIAR Research Program on Climate Change, Agriculture, and Food Security (CCAFS) proposed that this transformation should focus on four crucial action areas [16]: (1) rerouting rural farming and rural livelihoods to new sustainable, equitable trajectories; (2) de-risking livelihoods, farms, and value chains from extreme climatic events, including the implementation of DEWS; (3) reducing food system emissions; (4) realigning policies and finance to arrive at sustainable food systems [16]. However, this and many related initiatives to transform agri-food systems are actually not on track as per their own stated timelines [15,16]. To transform agriculture on-farm, efforts are needed to understand and address the observation that farmers have different starting points and experience different trends, and to establish how to enable them to reach sustainable, nutritious, equitable, and ethical new food systems. Small farm size and relative inefficiencies may favor farm consolidation to medium-sized holdings in certain cases [20,21]. Some farmers may wish to leave farming, while eager, well-educated others will be excited to enter farming, in ways that are intellectually challenging and financially rewarding [20,21]. This development should lead to a more modern, science-savvy leadership in agriculture, taking up modern technologies, including involvement in DEWS development and uptake [20].

With regard to agri-food system transformation, advances in digital technologies that combine remote sensing (RS), geographic information systems (GIS), and ICT-enabled citizen science can facilitate precise mapping of current assessments, and monitoring of drought trends from their early onset forward. Drought early warning and monitoring are crucial components of drought preparedness and mitigation plans [22]. The remote sensing literature offers numerous examples proposing earth observation techniques to support assessment of drought conditions [23]. Increasing access to open data, high-resolution remote sensors, and enhanced computing facilities have led to a new set of sophisticated techniques for DEWS in agriculture experiencing drought stress conditions [24], which monitor evapotranspiration [25], soil moisture [26], ground water fluxes [26,27], and precipitation.

Digital approaches can identify drought hazard hotspots and their dynamics and prognosis. In addition, new technologies can help facilitate the response to drought, enabling new farming systems that will reduce drought hazard impacts and even restore landscapes [28]. At the same time, new technologies can identify and help communicate risks to farming communities and other agri-food system stakeholders. However, many technological innovations, including improved climate forecasts, still need to reach their full positive impact in transformed agri-food systems [29], and finance for the transformation of land use related to agri-food systems received only around 1% of all climate finance in 2016 [30].

The Covid-19 pandemic inspired communities in challenged food systems to explore alternatives, such as new input sources, alternative local markets, more resilient inventory management, reviewing staff safety practices, monitoring the progress of planting and harvest, and implementing more flexible human resource management as absenteeism rose. In the long term, these may contribute to transformation of food systems once the pandemic subsides [18,31].

## 4. Drought, a Multiscale Phenomenon

Drought progresses through phases of reduced rainfall, water shortages, lower food production, food price spikes, reduced food security, increased hunger and outmigration, and ultimately famine. Drought resulting in famine is probably the worst environmental disaster. Here, we address two different scales at which droughts operate, global and local, and the consequences of inaction.

## 5. Drought as a Global Phenomenon

Droughts can cover anywhere from a few square kilometers to large sections of continents, and they can last from a few weeks to decades [32]. Droughts often have a slow, creeping onset, with incremental long-term impacts. This slow start may delay initial responses, which leads to situations in which thresholds for reversibility are exceeded. In spatial and temporal terms, paleoclimate records show that, across North America, West Africa, and East Asia, this has resulted in droughts that initially affect a small area and then grow to affect millions of square kilometers, potentially lasting from a single season to years and even decades [1]. As they grow in intensity, about 10% of droughts may travel several thousand kilometers from their original region, as in the western USA, which will be key to predictive DEWSs [33].

Because many droughts take an extended period to build up and are often forgotten once over, it has been difficult for scientists, natural resource managers, and policymakers to develop plans and take action for drought preparedness, active risk management, monitoring, DEWSs, impact assessment, coping ability, emergency, and long-term response and mitigation [34]. The WMO and GWP (2014) developed guidelines that include 10 generic steps for a National Drought Management Policy Template for Action, but uptake and implementation has been slow, with crisis management often being the default response to drought [34].

In addition to reducing the amount of available water, drought also affects water quality, which in turn negatively impacts ecosystems, biodiversity, including pollinators, land degradation, salinization of water bodies and groundwater, toxic blooms of algae and cyanobacteria, higher water turbidity, and consequent reduction in dissolved oxygen [35]. Thus, droughts affect sectors beyond only agriculture, including environmental, economic, and social domains [34]. Such diverse impacts include interrupted hydro-energy production, reduced food security, increased gender inequalities, compromised health, poverty, famine, conflict and civil unrest, and outmigration [35]. The complexity of so many direct and indirect costs in such diverse sectors makes it difficult to estimate total costs due to droughts and, equally, the benefits of counteraction [35].

The most drought-prone regions in the world from 1981 to 2010 were the Amazon Forest, the Mediterranean region, Africa, northeastern China, southeastern Australia, northwestern USA, and southwestern Canada. Most of these regions are also projected to be hardest hit by drought in the coming period 2071–2100 [36]. Droughts are predicted to increase in many parts of the world, but with diverse impact. For example, North Africa will experience greater drought hazard but a low risk ratio because of relatively advanced socioeconomic sectors that stabilize population growth and reduce drought vulnerability, while Central Africa will see the highest drought risk ratio, because of their vulnerability and relatively large population growth [37]. In a study of the drought vulnerability of the economic, energy, infrastructure and health sectors, land use, society (human development index, unemployment, net migration), and water resources in 46 African countries

during 1960–2015, the least vulnerable countries would remain as such into 2020–2100, and the most vulnerable countries based on historical data were predicted to become even more vulnerable [38].

In recent years, interest has grown in drought types that are sub-seasonal, have a sudden onset and rapid intensification, and continue over a matter of weeks or months [32]. High temperatures and low humidity, which result in high evaporative demand when precipitation is lower than average, can cause such quick depletion of soil moisture and drought, negatively affecting vegetation [39]. They can even occur in densely cropped humid and semi-humid regions, where water extraction from the soil is already high, as identified in China, where flash droughts more than doubled in frequency between 1979 and 2010 [40]. Research is still in its infancy, and even a definition of these flash droughts remains to be agreed upon by the stakeholder community [41]. Pendergrass et al. [32] (p. 195) proposed the following definition: "50% increase in EDDI (Evaporative Demand Drought Index) (toward drying) over 2 weeks for international situations, or a two-category change in the USDM (US Drought Monitor) for the USA in 2 weeks, sustained for at least another 2 weeks". Otkin et al. (2015) suggested that the rapid rate of intensification compared to other drought types during the development phase should be the basis for defining flash droughts [42]. This definition also allows for flash droughts to transition into one or more of the other drought types mentioned. The metrics required were presented by Otkin et al. (2021) in the form of the flash drought intensity index (FDII), with validation using data from the 2012 flash drought of highly cropped land in central USA [43]. Other metrics are also being developed, such as the probabilistic and multivariable flash drought identification method (PMFDI), based on both sudden onset and rapid intensification [44]. While much about the development of flash droughts remains unclear, it is clear that DEWSs capable of providing accurate and timely current assessment information on ongoing conditions are instrumental in rapidly tracking and responding to such droughts. Accurate forecasts on that short time scale are challenging, and they will need to address flash drought onset, intensification, and duration, along with associated temperature, humidity, and wind. More historical research of past flash droughts is needed, as is the incorporation of flash droughts into models of climate change [32]. Nevertheless, some models already project an increase in exposure risk to human-induced flash droughts in some countries such as China, even in traditionally more humid regions [45].

## 6. Droughts and Local Consequences

Drought risk to human communities is determined by the magnitude of the hazard, the level of vulnerability of the community, and the extent to which the community is exposed to the hazard [6]. Droughts create synchronous challenges over temporal and spatial scales, affecting physical and socioeconomic processes. Drought affects all four components of food security: availability, quality, access, and stability. Drought reduces crop and animal production and, potentially, quality, with greatest impact in rainfed agriculture and for subsistence farming households. For example, in the northern region of Minas Gerais in Brazil, large farm properties represent less than 5% of all rural properties, but occupy about 40% of the total farming area, while small farm properties (of about 20 ha each) represent 80% of all farm holdings. Between 2012 and 2017 these smaller farms of mostly socially vulnerable subsistence farmers represented 70% of the holdings affected by repeated droughts [46]. Subsistence farmers in the developing world suffer most from droughts at the local, individual farm level and as communities [47]. If groundwater levels drop to unsustainable levels, entire ecosystems can change, creating land degradation and compromising flora and fauna, possibly for a long period. In river networks, there will be different effects on upstream and downstream communities. Decisions on the allocation of limited water resources can create friction between local agricultural and urban communities, with tradeoffs to be considered. Even within agriculture, crops and animals may have different economic value, with tradeoffs again on where to apply restricted water

reserves. Food prices may rise as a result of reduced food production and imports, with the latter raising issues around global virtual water trade. In severe cases, entire communities may be uprooted directly or indirectly as irrigation basins and dams are developed to store water and mitigate drought-related problems in the future. Many kinds of policies are needed to address drought avoidance, reduction, or reversal, with local and international food transport and trade and social safety nets important immediate responses [48].

We can distinguish four types of droughts [6,49]:

1.  Meteorological drought: a prolonged reduction in precipitation relative to past average amounts, increased temperature, decreased humidity, and increased evapotranspiration.
2.  Hydrological drought: the availability of surface and subsurface water resources, including water storage and flow (e.g., lakes, streams, aquifers), being reduced due to lack of precipitation, a substantial deficit in surface runoff relative to past average amounts, or ground water not being recharged.
3.  Agricultural drought: soil moisture falling below crop requirements, leading to wilting crops and reduced yields.
4.  Socio-economic drought: human activities being disturbed, economic losses incurred, due to meteorological, hydrological, and agricultural droughts, and reduced water availability, which result in socioeconomic impacts, and may lead to human famine and starvation.

While our focus here is on agriculture, to understand and predict the impact of drought on agriculture requires attention to all types of droughts in order to arrive at an integrated way forward. A fifth type, environmental drought impacting natural resources, is increasingly studied, not least because ecosystems often express unique strong resistance and resilience to droughts, from which we can learn. However, more negative effects of structural environmental droughts may occur under future climate change [50]. Tarnavsky and Bonifacio [51] object to classifying droughts as "types", but rather consider them different "aspects" of drought, because similar monitoring methods (e.g., rainfall deficit) are used across the types. In addition, more than one type of drought often occurs during any drought event, and these often follow in sequence the order of the four types listed [39,51]. This implies that mitigating drought's first meteorological manifestation can mitigate or even prevent subsequent types. Droughts that affect local agri-food systems may also have impacts at national and regional scales, affecting food security, human and animal health, livelihood security, and personal security, each of which requires specific indicators [52].

DEWSs often focus mostly on infrastructure and technology, leaving the active engagement of communities and other vulnerable groups relatively underattended, even though they aim to address the DEWS objectives of risk knowledge, monitoring, dissemination, communication, and response capability [53]. Pulwarty and Sivakumar [1] reviewed more than 20 DEWSs, and they confirmed that links among local community-based, regional, national, and international approaches are often weak.

## 7. The Consequences of No Action

In some cases, no action is taken after a drought early warning. In others, action is incomplete. Almost 20 years after the EU approved the Water Framework Directive (WFD) in 2000, the vulnerability (defined by the IPCC as exposure, sensitivity, and adaptive capacity) of some European countries was still very high, with emergency plans in place in just 8.5% of the cities mandated to do so by 2007 [54]. Thus, even with legislation to prepare for drought, it is common for little or no action to be taken [54]. This is despite the general observation that the cost of strategic preparation is considerably lower than the cost of response and recovery following inaction [36].

In the USA, during stakeholder meetings, ranchers indicated that several weeks advance notice of emerging drought would allow them to take appropriate action, such as

destocking drought-sensitive pastures or purchasing supplemental feed [42]. Farmers also expressed interest in drought forecasting information, as it would help them make timely market decisions [42] and prepare insurance claims [55]. However, during the 2016 flash drought in the Northern Great Plains, range and livestock managers did not actually respond to early warning information about drought emergence and development from various sources, including news media and online. Managers decided to destock pastures only when they observed actual, environmental drought impacts, several months after the drought had started and too late to avoid serious losses in range productivity, diversity, and health. Clearly, many managers did not see early warning information as a call to action [56].

Nhamo et al. (2019) argued that, despite several DEWSs being available and used in the 15 countries of southern Africa, which experiences frequently recurring droughts, none contain a true regional drought forecasting system. Mostly, they only monitor and assess drought, thus preventing the formation of resilient communities and limiting action to post-disaster relief [49].

Meissner and Jacobs-Mata (2016) pointed out that, although models predicted El Niño-induced droughts for South Africa in 2015, instead of responding with early preventative measures, only remedial interventions were belatedly implemented [57]. All but one of South Africa's provinces were declared disaster areas due to the drought; yet, in a survey of cattle farmers in North West Province, 84% said that they had never received DEW information. Only 51% had enough grazing for a season under the drought, 20% rated the drinking water for their cattle as not dependable, 84% did not have enough fodder reserves or even fodder storage facilities, and 56% said they had no drought mitigation plan [58]. However, 95% said they had had contact with agricultural extension agents, which makes it even more surprising that the farmers were not made aware of the importance of planning ahead for drought risks [58]. Resolving the lack of agricultural extension messaging is as important for the rollout of DEWS as developing a functional system that addresses user needs. The literature is full of indicators, indices, and DEWS models, but short on successful rollouts.

Another survey, of farmers in Matabeleland North Province in Zimbabwe, where drought is frequent and land continues to degrade, uncovered more reasons for unsuccessful DEWS rollouts [59]. Two-thirds of the population had only primary education, which the authors considered a hurdle in conveying complex agricultural innovations, disaster preparedness, or the concept of climate change. However, although farmers reported that they noticed that rainfall patterns were no longer consistent, most took no adaptive action. Many (77%) received training in one or more agricultural practices, such as conservation agriculture, but not a single respondent confirmed awareness of drought preparedness policies that were in place since 1989. The researchers came to understand that no community leaders were included in the drought preparedness planning process. Farmers said they got drought early warning information from three sources: indigenous knowledge (46% of farmers), extension officers (33%), and radio and television (17%). Belle et al. (2017) concluded that enhancing drought preparedness and DEWSs should include both indigenous and science-based information [59].

Brazil has a long history of drought preparedness dating back to initiatives taken after the severe 1877–1879 drought, with the first institute to focus on water distribution, the Inspection Agency for Works against Drought, established in 1909. An analysis identified more than 15 different national and state agencies responsible for weather and climate forecasting, agrometeorological information, and related research, in addition to numerous municipal agencies. However, institutional coordination, integration, and articulation of responsibilities, including on DEWSs, remain nonoptimal. Overall response to droughts remains reactive rather than proactive and strategic, with existing infrastructures underused as a result of the lack of joint planning. This is despite Brazil's considerable expertise in the scientific and technical aspects of meteorology, climatology, and agricultural and hydrological monitoring and forecasting. There is a systemic lack of

institutional coordination and integration of high-quality know-how in drought vulnerability, resilience, monitoring, forecasting, early warning systems, impact assessment, mitigation, and response planning [60]. Brazil is a good example of the situation in many emerging intermediate countries, which have a good knowledge base and expertise but poor implementation of drought control strategies [61].

Lack of action, even where awareness seems to be present at least on paper, is spawning initiatives such as the "How to Communicate Drought: A guide by the Integrated Drought Management Program in Central and Eastern Europe" to spread the message and increase trust in the messenger. The guide points out that, with new modes of communication multiplying and different population segments using different channels, new tailor-made solutions may be needed [9].

## 8. Key Aspects of DEWS

Here, we discuss a selection of DEWS, chosen to address key aspects in their development and successful implementation, from which others can learn.

## 9. Global

The Global Drought Information System (GDIS), within the USA National Oceanic and Atmospheric Administration (NOAA, Washington, DC, USA), joins national and local partners to share their drought information and allow comparisons (see https://gdis-noaa.hub.arcgis.com/, accessed on 1 June 2021). The Global Drought Monitor is one of their systems (https://gdis-noaa.hub.arcgis.com/pages/drought-monitoring, accessed on 1 June 2021). Forecasting information, mostly from advanced countries, is integrated into the Global Data-Processing and Forecasting System in the form of online interactive maps. New generations of NOAA satellites are launched regularly. The 2017 fleet carries the Visible Infrared Imaging Radiometer Suite (VIIRS) sensor, which enables prediction of drought intensity, duration, and agricultural production loss 2 months in advance at 0.5 km$^2$ resolution [4].

The FAO operates the Global Information and Early Warning System on Food and Agriculture (GIEWS) (www.fao.org/giews, accessed on 1 June 2021), which was established after the food crises in the 1970s [62]. It encourages regional and national initiatives for DEWSs. FAO also developed the Agricultural Stress Index System (ASIS), using the Vegetation Health Index (VHI), to monitor water deficits and droughts [62,63].

The global Famine Early Warning Systems Network (FEWS NET) contains the food security-related Drought Early Warning System (DEWS), which allows for rapid identification of the location, extent, severity, and causes of sudden food security risks due to drought and has been under continuous development for the past 35 years [2]. Various organizations contribute, including FAO's GIEWS, the World Food Program Humanitarian Early Warning Service, the WMO/UN Strategy for International Disaster Reduction, and the Famine Early Warning System [64]. Expert judgement by regional analysts on-site around the world, making use of the main data monitoring and modeling portals, supports the decision-making process and has helped refine the tool [2]. Scenarios are developed on the basis of preliminary climatological assumptions derived from historical climatology (e.g., precipitation) or from models of past drought events, such as the El Niño-Southern Oscillation. Probabilities are known for drought events and the likelihood of relative food insecurities, known as the Food Security Outlook (FSO), before the cropping season. As the crop season unfolds, fresh monitoring data are continuously uploaded to update the FSO and guide early warning communication. The FEWS NET Land Data Assimilation System monitors current non-precipitation hydrologic variables, such as run-off, soil moisture, and water stress, comparing them with historical records. FEWS NET's correct predictions enabled timely action and expedited the targeted disbursement of more than 300 million USD in humanitarian aid to those struck by severe droughts in 2016/2017 [2].

## 10. Regional

Building resilience against regional droughts requires joint action among multiple centers of decision making, responsibilities, and knowledge [65]. All involved can be better prepared for future droughts by learning to resist shocks, adapt to change, and transform systems, but unfortunately there are few functional examples of regional DEWSs.

The North American Drought Monitor (NADM) is a collaborative DEWS involving Canada, the USA, and Mexico, initiated in 2002, which allows continent-wide monitoring and assessment [66]. It is based on the USDM, and it assesses and communicates the state of drought on a weekly basis.

The Economic and Social Commission for Western Asia notes under Regional Technical Challenges that the Arab world, mostly located in semiarid and arid regions, has failed to develop a regional, people-centered DEWS [67]. ESCWA calls for the necessary elements to be regionally developed, with advances being made in the African Drought Monitor and the Regional Drought Management System for Middle East and North Africa (MENA RDMS) [68]. Drought monitoring systems in MENA have been criticized for a narrow focus on precipitation data rather than composite drought indices integrated into a proper DEWS, and for lacking wider stakeholder involvement [69].

A positive example of regional cooperation, established after the severe 2015–2016 drought in Papua New Guinea (PNG), which affected 40% of its population, is the regional Climate Risk and Early Warning Systems (CREWS), a collaboration involving the PNG National Weather Services, the Australian Bureau of Meteorology, and WMO. CREWS predicts rainfall with a 3 month lead time, and the partners hope to expand the initiative across the Asia-Pacific region [70].

In Europe, about 11% of the population is considered at risk due to drought [71]. The EU established its Water Framework Directive (2000/60/EC) in 2000, with a subsequent nonbinding "Drought Management Plan Report Including Agricultural, Drought Indicators, and Climate Change Aspects" in 2007. The European Drought Observatory (EDO) was established in 2012, providing drought indicators and functioning as a DEWS [72]. By 2019, 78 River Basin Districts, representing 42% of all such districts in the EU, developed Drought Management Plans [72]. Such regional projects require regular improvements as partners and technologies evolve. EDO currently reports soil water droughts once a week, looking 3 months ahead at moisture anomalies. Nevertheless, more detailed hydrological predictions (e.g., stream flows, groundwater) are often required an entire crop season ahead, especially in light of recent increases in the frequency of drought events in the EU [73]. To address this need, a new DEWS was implemented in 2018. The ANYWHERE DEWS (AD-EWS) is funded by the EU Enhancing Emergency Management and Response to Extreme Weather and Climate Events (ANYWHERE) project and provides seasonal drought indices on precipitation, soil moisture, runoff, discharge, and groundwater, at higher spatial resolution (5 km × 5 km) than EDO. AD-EWS projects the start and end of soil moisture drought events up to 7 months in advance, on the basis of which users can take contingency actions, such as crop irrigation or ecosystem water conservation [73].

## 11. National

DEWSs have been in place for decades in some countries, such as Canada, where a Forage Drought Early Warning System (FoDEWS) has been operative since 1982 to manage cattle grazing on prairie pastures. The model calculates soil moisture from weather records relative to historical data to predict subsequent droughts [74].

The USDM, launched in 1999, is one of the oldest and most reliable DEWSs and is linked to the US Department of Agriculture (USDA) emergency disaster interventions, making it most relevant for our emphasis on agri-food systems. It is produced jointly by the National Drought Mitigation Center at the University of Nebraska-Lincoln, NOAA, and the USDA, and provides weekly status assessments. In a recent study over 2001–2013,

USDM, applying GIS-enabled information on agricultural areas, was able to identify significant negative impacts of drought on maize and soybean yields, beyond the effects of precipitation and temperature, thus complementing weather information [75]. Recently, a probabilistic methodology was developed for a non-discrete USDM index with dryness/wetness value on a continuum, which will facilitate USDM forecasting methods, exemplifying the importance of continuous improvement of tools [76].

In Queensland, Australia, hybridization of the adaptive neuro-fuzzy inference system (ANFIS) with other machine-learning, metaheuristic optimization algorithms significantly increased prediction accuracy of spatial occurrence and distribution of drought over ANFIS alone [77]. Plant-available water capacity, percentage of sand in soil, and mean annual precipitation were the most valuable drought-predicting factors [77]. With NOAA's VIIRS-based vegetation health (VH) technology to determine VH, while wheat is most sensitive to drought, it was possible to create early warning of drought-related wheat yield losses and predicted yield 1 to 2 months ahead of harvest [78]. An artificial neural network trained using drought information in the period 1901–2010 and tested for 2011–2018 predicted temporal drought trends in New South Wales, Australia with a coefficient of determination ($r^2$) of 0.86 [79]. The Temperature Rise Index (TRI) proved promising for agricultural drought early warning in Australia's wheatbelt in 2015–2019, strongly correlated with wheat yield ($r > 0.8$) and with a 1 month lead time over the Vegetation Condition Index [80].

Understanding patterns, trends, interactions, and feedback among different variables in complex systems (including, for example, hydrology, economics, social sciences, and laws) is facilitated by applying a system dynamics modeling approach, which was used to develop a decision support tool for water management of the Jucar River system in eastern Spain [81]. This tool included early warning system components, which provided accurate quantitative outputs compared to historical records. Such system dynamics modeling could be explored more, including for studying tradeoffs [81].

As indicated above, developing DEWSs for flash droughts is particularly challenging because of their rapid onset; nevertheless, such models are urgently needed. Ford et al. (2015) looked at data from 2000–2013 in Oklahoma, USA to compare the effectiveness of the USDM with in situ soil moisture observations [82]. They found that soil moisture percentiles, derived from sensor-derived volumetric water content calculated hourly, provided a 2–3 week lead time. In the 2017 flash drought in the US Northern Great Plains and the adjacent Canadian prairies, Hoell et al. (2020) report that, in 1 month in late spring, soil moisture dropped from the 80th percentile to the 15th percentile. Impacts were multisectoral, including widespread wildfires, poor air quality, damaged ecosystems, degraded mental health, and diminished tourism and recreation. In such sparsely populated areas, a more systematic network of automated and remotely sensed observations of key weather and hydrological variables is needed [83].

In Ethiopia, a DEWS was developed in 2008 that includes drought assessments coming from the Early Warning and Response Directorate (EWRD) with the Livelihoods, Early Assessment and Protection (LEAP) tool. LEAP estimates the number of people needing food assistance during the two main cropping seasons in Ethiopia on the basis of crop, rainfall, and water requirement data, allowing it to compute estimates of crop yield reduction under drought. As a result, the Productive Safety Net Program (PSNP) was able to rapidly scale up the provision of resources to the 7.6 million beneficiaries threatened with drought, reducing response time to one-quarter of what is was before. This DEWS approach saved Ethiopia from the worst impacts of the Horn of Africa 2011 drought, which hit its neighbors severely [84].

"Preparedness at all levels to respond to the warnings received" is an integral part of the definition of DEWS [9]. Slow response to early warning of the Horn of Africa drought in 2011 and subsequent famine was a system-wide failure. In August 2010, for example, FEWS NET started issuing accurate and timely warnings of a severe oncoming drought; a year later, the declaration of famine in July 2011 prompted scaling up the aid response.

As a result, an early-warning, early-action trigger mechanism was established in Somalia, the Early Warning, Early Action (EWEA). The explicit "early action" in the mechanism's name emphasizes that a timely response it crucial. To achieve that, an accountability framework was developed, which identifies the people in the humanitarian community who should carry out pre-agreed responsibilities within strict time schedules, to achieve the rapid response. In 2016/2017, the EWEA underwent its first real test during a drought, before it had been officially institutionalized, and areas were identified where the accountability framework needed improvement. One question, for example, was whether to hardwire the funding of actions that would be triggered [85]. The African Risk Capacity (ARC), part of the African Union, did just that, and it has paid out 61 million USD for preapproved contingency plans by the country as early response aid, through innovative financial mechanisms such as pooled risks and index-based insurance [86]. Following meteorological developments in near to real time as they impact vulnerable populations allows for rapid targeted disbursal of relief funds, triggered automatically when preset risk profile thresholds are exceeded.

Ewbank et al. (2019) compared communities in Nicaragua and Ethiopia during the 2015–2016 El Niño drought. Some communities received drought early warnings, while other communities were informed later or not at all. In both countries, those receiving early warnings took one or more early drought-resilient actions, such as adapting planting time, land preparation, crops to be grown, crop varieties, fertilizer use, and water conservation. These steps resulted in clear advantages in yield and input use efficiencies for certain crops in Nicaragua for those responding to DEWS, and a greater conviction by farmers that they avoided major crop damage. Farmers who received DEWs in Nicaragua responded more successfully than those in Ethiopia, probably because DEWS measures were advocated during community-based resilience-building in Nicaragua in the preceding 5–6 years, but only for 2 years in Ethiopia. Clearly, long-term local follow-up with awareness raising, training of effective responses to DEWSs, and long-term resilience-building are needed to translate warnings into positive impact [87].

Implementation of a DEWS through training and information reduced the threat of food insecurity by 24% and increased household nutrition by 30% in the agro-pastoralist Karamoja subregion of Uganda, which experiences recurrent droughts [88]. The intervention used a farming systems approach that included recording data and advising on time of planting, diversifying crops (e.g., groundnuts, maize, pearl millet, sorghum, sunflower, wild vegetables, and wild fruit), animals (e.g., goats, sheep, camels, chickens, and fish), honey, mechanization, and improved, drought-tolerant varieties. Beneficiary households ate more meals per day with more diverse food, and experienced lower household food insecurity than control households that did not receive the DEWS training. This encouraging study created an interest in implementing DEWS across the entire country [88].

In Kenya, Barrett et al. (2020) used a DEWS to forecast drought stress in vegetation targeted for livestock grazing in rangelands. A 3 month averaged Normalized Difference Vegetation Index (NDVI)-based Vegetation Condition Index (VCI3M), which is strongly linked to agricultural production, was already being used to establish various drought-related parameters, but not in forecasting. VCI3M values below 35, an alert marker that indicates a state of drought, trigger government action. Machine-learning approaches were used to predict VCI3M values and when they would drop below the drought alert marker. The aim was to allow sufficient lead time (i.e., weeks) to implement drought preparedness measures at a high level of prediction accuracy. Linear auto regression and Gaussian process modeling methods were able to predict VCI3M values dropping below a threshold of 4 weeks into the future at about 89% probability, thus allowing for timely action to limit serious drought impact [89]. In Shaanxi province, China, machine learning further enhanced accuracy in forecasting droughts up to 6 months ahead, with accuracies above 90% [90].

In Iran, a statistical downscaling model using Palmer Drought Severity Index values from 1995 to 2014 predicted decreasing precipitation and increasing temperature for 2019–

2048 in Fars province with coefficients of determination of 0.63 and 0.95, respectively, between simulated and observed data [91]. Sharafi et al. (2021) studied farmer decision making on adoption of a DEWS in one region of Iran to understand why uptake was lower than expected. Farmers reported three major considerations: information being from valid sources, information that increases their income, and information provided in simple language. Thus, not only maximizing profit, but also socioeconomic and sociopsychological considerations play a role in farmers' adoption of DEWS. As their study was of limited scope, the authors stressed that more research of rainfed farming households is needed [92].

## 12. Dimensions of Drought Early Warning

Drought early warning involves responding not only to meteorological and environmental factors, but also to detrimental drought-provoking activities by humans, as well as their role in biodiversity and crop diversification. Often, these are not taken into account, including in DEWS development, with its greater emphasis on technical aspects. We outline here some of the dimensions of these three domains, which can contribute to drought initiation and intensification. These domains are worthy of attention because, in all three areas, we can take relatively quick measures to adjust our own behavior, which can help lessen the onset and development of droughts. We advocate for DEWS development to take these areas into account, not least by including all stakeholders involved, a point we return to where we discuss the importance of citizen input and extracting worthwhile information for DEWS from indigenous knowledge sources.

## 13. Role of Human Activities

In the Anthropocene, interactions that involve meteorological events and land use need to be considered alongside human activities, which can also be very detrimental and contribute to drought, requiring new definitions of some drought concepts and societal adaptation [93]. Humans may be responsible for excess water extraction, land degradation, and desertification, all of which can play a part in droughts [94].

One straightforward impact of human activities is extracting more from surface water sources than is replenished by natural precipitation and runoff. Recent population and socioeconomic growth in Iran are the drivers behind much increased extraction, especially for agriculture, creating water stress, and reflecting serious climate change impact [95]. Combined with decreasing precipitation, this anthropogenic water stress is projected to affect most water basins in Iran before the end of this century [95]. In a study of droughts in five major river basins in South Korea for 1973–2017, precipitation was the most important factor, and its absence increased drought severity by 41%, but human-initiated streamflow increased drought severity by 7–25% and duration by 5–28% [96].

Land degradation is another effect of human activity. Net primary production has dropped in almost one-quarter of total global land area, and anthropogenic degradation is extending into ever less fertile land [97]. Improving degraded land is possible, but requires long-term investment in enhancing sustainable intensification, including integrated soil fertility management and introduction of biomass and nutrients. However, if such remedial action is not taken, crop production decreases and soil organic matter (SOM) content declines further. As a result, formerly productive lands that are not restored undergo further deterioration and depletion, leading to increased vulnerability to droughts and even desertification. The value of SOM was quantified in one study of SOM and maize yields in 12,376 county-years in the USA. It showed that 1% of additional SOM added 2.2 t/ha under severe drought. The benefit is partly explained by increased available water and cation exchange capacity [98].

## 14. Role of Biodiversity and Crop Diversification

Drought often results in biodiversity loss, but equally important is how increased biodiversity can protect ecosystems from negative impacts of drought in the first place.

Biodiversity (the number of species within a fixed area) reduced the negative effect of drought on household incomes in a study of 7556 agricultural households in 23 developing countries, where timber and non-timber forest products, including bush meat, provide additional income of about 20% [99]. A survey of more than 20 crops in 311 districts in India from 1966–2011 showed that prior crop diversification reduced the impact of water deficits [100]. In Nebraska, USA, a comparison of maize in monocultures with maize in rotation with 2–5 cover crops of cereals and legumes over 16 to 58 years showed that maize in rotation outyielded monocultures by about 23%, and, under drought yield loss in monocultures, it was 14–90% higher [101].

Grassland experiments in North America and Europe, in which the response to reduced water availability was studied for a range of biodiversity richness, showed greater yields with higher biodiversity in drought and non-drought treatments [102]. In Switzerland, trials over 7 years with up to 60 different species in grasslands demonstrated that more biodiverse grasslands experienced less drought stress than less diverse ones. Greater diversity was also associated with greater resilience to stress [103]. Grassland biodiversity experiments across Europe, USA, and Canada showed that high-diversity plant communities were more resistant to even extreme, prolonged droughts, maintaining ecosystem productivity [104,105]. The exact mechanisms via which biodiversity protects against drought are not well known, but one 10 year controlled study in the Netherlands, which included two severe summer droughts, concluded that the presence of drought-tolerant species in the mix alters and ameliorates dry conditions as an emergent property of biodiversity, while also improving the performance of drought-sensitive species when water is limited [106]. A meta-analysis of 56 articles on agroecology and human food security and nutrition in mostly farming households in low- and middle-income countries found that 78% showed a positive contribution of sustainable, agroecological practices to the Household Food Insecurity Access Scale and dietary diversity in households. Such practices included crop diversification, intercropping, agroforestry, integrating crop and livestock, soil management measures, and farmer-to-farmer networks, with trends indicating that adopting more of these practices further increased food security and nutrition [107].

## 15. Incorporating Participatory, Local, and Indigenous Knowledge into DEWSs through Citizen Science and Enabling ICTs

Broadening the involvement of people on the ground and including closer consideration of their agri-food system practices, some of which were discussed above, can improve the development and implementation of DEWSs. A drought monitoring study of four countries in the Near East and North Africa used a participatory approach to assess drought monitoring needs, reflecting the fact that drought impacts not only agriculture, but also broader environmental and social systems [69]. Interviews, focus groups, and workshops were organized with farmers, government agencies, civil society organizations, the private sector, and research institutions, during which participants ranked the importance of hydrological, agricultural, ecological, and socioeconomic drought impacts. Modern climatic, hydrological, and vegetative condition monitoring approaches were widely discussed, along with the most important agricultural, ecological, and socioeconomic drought monitoring indicators. Thus, the broad stakeholder base identified the top priority drought monitoring needs to improve the DEWS [69]. These were as follows:

1. Drought definitions. The technical definitions of drought should move beyond just precipitation deficit, to increase rigorous monitoring of all relevant factors, facilitate the drought declaration process, and allow staged intervention processes.
2. Information sharing. There is a need to formalize and automate data-sharing processes among all involved countries, with possibly a shared drought data platform.

3. Ground truthing of remote-sensing-derived data. In order to build on the remote sensing and modeled data, more ground truthing is needed to increase the acceptable levels of accuracy, precision, and geographic scale, so that such data reflect the real drought impacts that observers see on the ground.

4. Intersectoral engagement. Interactions among farming communities, the institutions that represent them, and government agencies responsible for drought management need strengthening. These include the opportunity for farmers and government agencies to exchange their own, often tailor-made, drought-related information both ways, as both parties have unique critical information on drought (potential) impacts.

These are examples in which regard other DEWSs can also be improved to develop wider participation and a broader knowledge base.

A further step in inclusiveness is to involve the agri-food system community from beginning to end, with community initiatives to collect drought risk information, prioritize dissemination of drought early warning messages among at-risk groups, and facilitate emergency response implementation [108,109]. Such an approach is a community process, as distinct from the often more linear design process of an early warning system. It requires keeping people and communities at the center of the entire process, with a strong education element. Implementing the DEWS becomes a social process, rather than experts "handing down" information to vulnerable communities [108,109]. An inclusive, polycentric approach to risk governance, linking remote rural drought-related processes with local, regional, and national processes, is gaining wider adoption [65]. Preparedness becomes part of daily practice, rather than a set of actions that are triggered only when a stress develops [108]. Implementation of early warning systems as social processes may require tailoring them for the specific needs of groups or communities, with appropriate communication and language tools for each setting [109].

In that context, it is noteworthy that, although mobile phone signals are within reach for 96% of the world's population, only one-third has experience searching the internet [110]. Furthermore, there are places where mobile phone possession is still very low, even lower than that of radios [111]. The Sendai Framework specifically stresses that simple and low-cost early warning equipment be tailormade through local participatory processes to increase adoption, and to include "indigenous peoples" and "older persons (who) have years of knowledge, skills, and wisdom" [7] (p. 23). One way to achieve the necessary greater participation on DEWS-related monitoring and adoption is through citizen science, linked by mobile-phone-facilitated communication and information sharing.

Citizen science is being applied in disaster risk reduction, although twice as commonly in developed countries than developing countries [112], which offers opportunities for mutual learning. Citizen science-based ICT, low-cost sensor networks, and developments in geospatial information sciences allow end-users a shift from well-meant but top-down assessments toward more community-based methods, which foster buy-in and empowerment [65]. Citizen science can not only facilitate preparedness through risk data capture and monitoring, but also can facilitate ground truthing, monitoring drought response and effectiveness, and subsequent post-disaster recovery and adaptation, thus building longer-term resilience, leading to reliable regional risk governance [65]. Citizen scientists also facilitate the community-wide uptake and appreciation of a DEWS strategy and culture, which permeates the society also in the periods when there are no droughts. Given that farming communities often live in remote areas, engaging those local farmers in data gathering can be a plus for those communities, government agencies responsible for DEWS implementation, and associated scientists interested in large but high-quality data gathering for subsequent analyses.

For example, concerns that coal seam gas activities in Queensland, Australia, would affect ground water and borehole water levels led to the establishment of the Groundwater Net and Groundwater Online platforms, a citizen science approach with more than 500 landholders to contribute information on groundwater level and pressure from their boreholes. This inclusive approach resulted in more frequent groundwater monitoring,

landholders being better informed through workshops, and landowners becoming more confident in dealing with private and public sectors [113]. Despite such inspiring examples, progress in actually engaging with communities, understanding their needs and constraints, and jointly implementing citizen science has been slow, with most efforts still mostly focused on technologies, methods, and systems [114,115].

DEWSs are not error-free and often contain aspects of subjectivity [116]. In the quest to make subjective experience a positive aspect of DEWS development, sourcing and including indigenous knowledge is gaining interest [53,116,117]. Farmers around the world have over time developed their own indigenous knowledge systems, based on observation of ecology, biodiversity, and climate interactions [116,117]. The inclusion of indigenous knowledge can increase early drought detection, as well as local relevance, trust, and uptake of DEWS [117]. Indicators include observations on flowering and fruiting of plants, water levels in streams and ponds, and animal behavior [116]. In Uganda, indigenous traditional knowledge provides inputs to early warning systems via observations of how birds, plants, insects, and animals respond to oncoming drought events, how wind directions shift, how day and night temperatures change, and how the color of the moon varies, allowing, for example, the prediction of seasonal rainfall [118]. DEWS data collection from these Ugandan households is carried out on a monthly basis at sentinel sites and is uploaded via mobile phones into a central system; after analysis, the data are reported as monthly early warning bulletins, although consistent incorporation in DEWSs has not yet been achieved [118].

Fuzzy logic facilitates the incorporation of indigenous knowledge into enhanced decision support tools, integrating it with modern scientific systems such as smart sensor technology and mobile phones. Complementary synergies can be captured from indigenous knowledge and modern science that are superior to either system in isolation [119]. Fuzzy cognitive mapping was used to understand and capture the individual and community perceptions of drought, impact on livelihoods, and concomitant coping strategies in a district in India's Telangana State [120]. Such an approach may help enhance the collective capacity for evidence-based decision making during future droughts. Subsequent modeling identified ecosystem-based adaptation measures and effective community-based management and governance [120]. The Rule-Based Drought Early Warning Expert System (RB-DEWES), developed in South Africa, is an AI-based expert system, designed to capture human expertise in a limited domain and capable of generating inferences, reasoning, and predictions, based on data. It includes indigenous expert drought knowledge, for example, on plant and animal behavior, weather phenomena, and astronomical indicators [116]. It contains reasoning techniques and a probabilistic attribution of confidence levels as to the certainty of the input, thus mimicking the indigenous drought expert with rule-set patterns of ecology–biodiversity climatic interactions [116]. It goes beyond this review and our knowledge base to discuss further how exactly these new science advances may be able to model and include citizen science and indigenous knowledge, but their promise certainly seems great.

## 16. Geo-Tagging and Agro-Tagging Inputs to DEWS

An appropriate early response to drought warnings requires ground validation, which may be supplied by citizen science or farmer community surveys, for timely submission of accurate, detailed information [85,121]. Geo-tagging the local drought area under consideration, with GPS-based surveys and geo-referenced field photos, can facilitate that by providing valid, location-based information [28,122]. Geo-tagged, time-stamped images provide high-resolution, visually interpretable data on crop growth stage and crop health that can complement remote-sensing data [123]. Triangulation, confirming information by comparing three independent sources, also facilitates collating relevant information [85]. Geo-tagged camera traps allow these devices to follow natural developments in the landscape before, during, and after stress, and the accompanying behavior of domestic and other animals of interest [124]. Farmer community surveys in Nigeria, Burkina

Faso, and Mali, using geo-tagged remote-sensing images, were used to evaluate the predictors of drought in 300 inland rice-based production systems and the factors affecting farmers' mitigation measures. These included the need to secure property rights, which increases farmers' willingness to invest in mitigation measures, with a focus on women's associations as they were found to be more likely to implement mitigation measures than men in the regions studied [121]. Agro-tagging, which involves adding additional information about on-farm activities to geo-tagged images, will be especially relevant in agri-food system settings to better understand the drivers for mitigation. Under droughts, geo-tagging and agro-tagging may help validate drought impact at the farm level and farming system level, and even provide a source of local validation of the DEWS.

## 17. Conclusions and the Way Forward

Drought early warning systems (DEWSs) have been implemented over at least the past 40 years. The rationale has been explained by local, regional, and international organizations. In essence, the costs of recovery from drought and rebuilding communities far exceed the investment needed to prepare for drought, including the implementation of DEWSs. Many countries currently operate either an international DEWS or a locally developed one, but many other countries, not only in the developing world, have yet to implement a DEWS, while others have not updated them. Given that droughts may deepen and expand if no action is taken, these consequences of inaction need to be discussed with slow adopters of DEWSs, as their inaction threatens neighbors and future generations. Agri-food systems, our emphasis here, involves including rural communities, which are often remotely located and poorly connected through modern infrastructure. Hence, we recommend that special efforts be made to include them in DEWSs from the development phase onward. In addition, we recommend that, on the basis of this stakeholder input, consideration is given to the development of new designs of DEWSs to better access remote data sources covering small farm-holding sizes with often very diverse cropping and animal husbandry patterns, in addition to current development for more densely populated, high-infrastructure regions, and for agricultural settings with large holdings and more monocultures.

The development of DEWSs has been driven first by greater computing power, followed by the identification of new drought indicators and indices, and most recently remote sensing using satellite-based technologies. This development has created a body of research that continues to grow. In particular, for remotely located agricultural settings, remote sensing that is able to distinguish agricultural and livelihood features and associated indicators within an individual small farm, which may not be larger than one hectare, is needed. This is a complex area of research as sensing down to such a low scale may also involve issues of personal privacy and even safety. We recommend that remote sensing of key indices of crop-diverse, smallholder farms relevant to monitoring the onset and development of drought in DEWS is increasingly studied and implemented. In addition, methods need to be developed to pass these findings and associated, often customized, recommendations back to the individual smallholders in a timely and user-friendly fashion.

Artificial intelligence and machine learning are now being explored to further fine-tune DEWS so that they provide more sensitive monitoring of current assessments and provide longer lead times for action to be taken. New indicators, indices, and DEWS are being developed, a healthy process of competition. However, it seems that less ground truthing is being carried out for some of the newer monitoring tools, probably because ground truthing is expensive and time-consuming, especially as remote sensing allows enlarging the potential target area. At the same time, we advocate studying small farms with relatively high levels of diversity. Nevertheless, some confirmation of the relevance, added value, and accuracy of new indicators is needed if they are to be included in new DEWSs. Citizen science can contribute to overcome limited ground truthing, especially linked to reliable geo-tagged observations. Particularly in smallholder farm settings of

small size, which may differ considerably despite being close together, farmers themselves becoming participants through citizen science monitoring will allow data on that rich diversity to be more fully captured and passed on to researchers for analysis and recommendations.

Droughts were previously seen and addressed as slow in onset; however, in recent years, flash droughts, which can have devastating impacts, have gained visibility. Current assessments of flash droughts can be well monitored by DEWSs, which are very valuable in deciding on measures to be taken; however, due to their speed, timeliness in response by communities and specialists will be a factor in enabling protective action.

Successes of global, regional, national, and local DEWSs are inspiring, but failures also provide lessons for the way forward. With regard to global efforts, we recommend that some global and regional organizations working on agri-food systems cooperate to identify successful local and regional DEWS applied successfully in agricultural zones from countries and promote them to similar countries that do not yet have effective DEWSs. While subsequent local adjustments will likely still be needed, at least a formal system will then be in place that can start to be built into a fully customized DEWS. We also recommend that, in the process, information on all aspects be freely shared among countries and organizations.

Growing interest in the socioeconomic impacts of drought has identified how human activities are interrupted by droughts and increase the sense of urgency. However, we also need to consider how some human activities may make drought more likely and more severe, for example, as a result of overextraction of water from reservoirs or groundwater, as well as land degradation due to unsustainable practices. Diversifying crops, trees, and animals on-farm and increasing biodiversity in ecosystems will also help build resilience in those agro-ecosystems, and we emphasize that these practices should be actively promoted as part of DEWS strategies.

It is clear that involving farming communities and others in agri-food systems in drought-prone areas in all aspects of DEWSs, from design to deployment to data collection, enriches the resulting system and facilitates its implementation, enhancing an appropriate response. We recommend that this path is followed especially where farms are small, diverse, and numerous, enhancing the active participant base. Local and indigenous knowledge on drought, which has been accumulated over generations by farmers, is being identified through more participatory and citizen-science approaches, and it is beginning to enrich some DEWSs. The use of science-based tools, including expert systems, fuzzy logic, and artificial intelligence, to capture indigenous, seemingly subjective, natural observations by local domain experts, is still in its infancy, and we recommend that these aspects receive more attention in dedicated research. ICT approaches are also reaching more remote regions and facilitating data collection and drought-relevant communication. As our emphasis here is on DEWS in agri-food systems, geo-tagging and agro-tagging are helping to link diverse sources of drought information, from hard science technology readings to videos of local plant or animal behavioral response in agricultural drought settings. Such integrated approaches will also help facilitate the ongoing agri-food system transformation.

Climate change models predict that droughts will increase in occurrence, intensity, duration, and spread, including in agricultural areas. Enhancing preparedness through DEWSs will enable monitoring of current assessments and timely, proactive action, thus avoiding the worst detrimental impacts of drought on human populations and the environment, which, in agri-food system settings, extend into threats to global food security.

**Author Contributions:** Original draft and writing, M.v.G.; Writing and review, C.B. All authors have read and agreed to the published version of the manuscript.

**Funding:** This research received no external funding

**Conflicts of Interest:** The authors declare no conflict of interest.

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
