# Peer review of "Drought Early Warning in Agri-Food Systems"

_climate, doi:10.3390/cli9090134_

Round 1
Reviewer 1 Report
This commentary addresses the role of Drought Early Warning (DEW) in agri-food systems and new advances that have taken place. The new advances identified include increased computing power, remote sensing technologies, artificial intelligence, machine learning, and fuzzy logic. The authors also address several dimensions of DEW including the role of human activities, role of biodiversity and crop diversification, indigenous knowledge, and geo-tagging and geo-logging inputs for the DEWS.
Overall, I thought that the topic for the commentary is valuable and that the commentary was well written. The mechanics were good throughout and there were very few edits identified, and these were mostly within the references. Given this, I think that the commentary provides a benefit for the readers of Climate. Based on my following comments, I recommend minor revisions for the article.
Let me begin by highlighting some of the strengths of the commentary. The first half of the document was very well done and I very much enjoyed the comprehensive presentation of the references related to the topic. I thought that Section 1.2 did a nice job identifying issues related with DEWS. I also liked that the authors included a Covid-19 perspective, which was a nice contemporary perspective.
Likewise, I thought that Section 2 was also well done. This included the description of the link between water quality and food systems, the discussion of the four components of food security, and the discussion of local vulnerabilities. I definitely agree with the emphasis on Extension services and the point made by the authors that there have been many indicators, indices, and DEWS described in the literature, but very few successful implementations of DEWS. This is a point that I will come back to below.
The following points are suggestions for how the commentary can be improved.
• As I see this commentary as an overview or review of DEWS for agri-food systems, I believe that previous reviews of DEWS should be emphasized to a greater extent within the text. Both references Funk et al. (2019, #11) and Pulwarty and Sivakumar (2014, #25) provide important reviews of DEWS. I think it would be nice right at the beginning in the Introduction for the authors to mention that these reviews exist and describe them…but then continue to say that this review for Climate is new and has a different perspective that builds from the other two reviews—providing additional value to the community. This particular commentary focuses on agri-food systems and the recent advances beyond what have been mentioned in references #11 and #25. I think this update would be an exciting upgrade for the commentary and help it stand out more in the community.
• The authors confuse predictions with early warning in a couple of places. Early warning can also include current assessments in the addition to predictions or outlooks. In one location, the last sentence of Section 1.1, the authors mention that multiple indicators and indices contributing to modern DEWS lead to greater uncertainty in the “final predictions”. This statement overlooks that DEWS can be composed of current assessments and/or predictions or outlooks. This text should be adjusted. Later, in Section 2.1, at the end of page 4, the authors mention that DEWS operating on a weekly or monthly timescale will be ineffective for flash droughts. I think it depends on the information, and current assessment information versus “forecasts” are huge benefits when trying to track rapidly evolving drought conditions. Thus, I disagree with the statement made by the authors. In fact, having accurate and timely assessments of the current conditions are the best ways that DEWS can utilize to respond effectively during flash drought events.
• Neither the U.S. Drought Monitor (USDM) nor the North American Drought Monitor (NADM) are mentioned in Section 3 for examples on the national or regional scales, respectively. These two tools are some of the oldest and most reliable DEWS available. The USDM has direct connections with USDA emergency disaster payments, making it an important tool for agri-food systems. Both the USDM and NADM also utilize many of the new products and technologies that the authors describe as important for their commentary. The two tools, for example, utilize GIS overlays to depict the drought impacts on important agricultural commodities across the continent in fine spatial detail. This text should be updated for sure!
• Although the Introduction talks about many of the recent advances, I feel that some of these are not well represented in the commentary. I would like the commentary to more directly address in more detail the increased computing power, remote sensing technologies, artificial intelligence, machine learning, and fuzzy logic topics mentioned in the Abstract. There should be a nice progression of these items discussed in the commentary. For example, the commentary mentions one tool that provides one perspective of drought (EDDI), but it does not describe the promising development of a variety of remote sensing technologies and tools. These advances get lost within the discussions in Section 4 (next two bullets).
• Similarly, in Section 4 on Dimensions, the section steps through the role of human activities, role of biodiversity and crop diversification, indigenous knowledge and citizen science, and geo-tagging and agro tagging. These are all great points, but I do not see their connection with the Introduction and the recent advances listed in the bullet above and from the Abstract. I think these could all be nicely connected and organized better to improve the commentary.
• Organizing both Section 4 and Section 5 around the above points would make both of these sections more concise and connected to the early sections of the article. Again, this will improve the commentary.
• I mentioned above how much I appreciated the authors comments regarding how there have been many indicators, indices, and DEWS described in the literature, but very few successful implementations of DEWS. I believe that addressing this more directly in the Conclusions, Section 5, would be valuable. The authors recap many of their points, but I was looking for a list (bullets perhaps?) of how DEWS can be improved for agri-food systems specifically. For example, design the section so that there is a recap of advances followed by recommendations for action so that the DEWS can be implemented.
• I think that the Section 3 title: “Existing DEWS and the pros and cons” could be improved. This section is not so much about the pros and cons, but rather about the “key aspects” of DEWS—as the text in the first sentence describes. Pros and cons misrepresents what is discussed in this section.
• There are two sections labeled as Section 4.1. The first is Section 4.1 Role of human activities. The second is Section 4.1 Role of biodiversity and crop diversification. Mark sure to renumber the sections appropriately.
• The formatting of the last paragraph of the Conclusions was strange and the wording was vague. I am not sure whether this is part of the commentary or not.
Reviewer 2 Report
Dear Authors,
The submitted manuscript is not concise, or precise. They are no objectives and it is difficult to follow ( even find out) the main idea. I would suggest the authors to: - make manuscript shorter, - put the title: DEW in Agri-Food System in the focus of their research - organize manuscript according to instructions - give concise conclusions and recommendations
Author Response
No response required by academic editor
Reviewer 3 Report
This paper attempts to deal with a very difficult issue which is drought monitoring and prediction systems. It is very crucial for the authors to clarify the terms “drought hazard”, “drought vulnerability” and “drought risk”.
For instance, drought vulnerability is expressed by the conditions determined by physical, social, economic, and environmental factors or processes which increase the susceptibility of a community to the impact of the hazard, including land degradation and desertification.
So, the authors should clarify/separate the drought early warning systems accordingly whether they address the hazard, the vulnerability, or the risk of drought.
Also, the authors should separate the drought early warning systems based on the different types of drought as well as the duration of droughts. Specifically, more information is needed about the early warning systems in case of flash droughts.
I strongly suggest for each category of the different drought early warning systems, their effectiveness to be identified (for instance in a summary table).
Round 2
Reviewer 3 Report
The manuscript has been improved upon revision. So, I recommend its publication.
Author Response
We thank Reviewer 3 for the suggestion and have made the requested change to the text. In addition, we are grateful to Reviewer 3 for noting that in our effort to renumber all the references in track changes mode, we introduced some additional carriage returns that made it seem as if 3 references were not correctly numbered. We have corrected this